# Can Glycosylation Mask the Detection of MHC Expressing p53 Peptides by T Cell Receptors?

**DOI:** 10.3390/biom11071056

**Published:** 2021-07-19

**Authors:** Thanh Binh Nguyen, David P. Lane, Chandra S. Verma

**Affiliations:** 1Division of Biomolecular Structure to Mechanism, Bioinformatics Institute, Agency for Science, Technology and Research (A*STAR), Singapore 138671, Singapore; nguyenbinhchem@gmail.com; 2p53 Laboratory, Agency for Science, Technology and Research (A*STAR), Singapore 138648, Singapore; dplane@imcb.a-star.edu.sg; 3School of Biological Sciences, College of Science, Nanyang Technological University, Singapore 637551, Singapore; 4Department of Biological Sciences, Faculty of Science, National University of Singapore, Singapore 117543, Singapore

**Keywords:** molecular dynamics, p53, HLA-A24, glycosylation

## Abstract

Proteins of the major histocompatibility complex (MHC) class I, or human leukocyte antigen (HLA) in humans interact with endogenous peptides and present them to T cell receptors (TCR), which in turn tune the immune system to recognize and discriminate between self and foreign (non-self) peptides. Of especial importance are peptides derived from tumor-associated antigens. T cells recognizing these peptides are found in cancer patients, but not in cancer-free individuals. What stimulates this recognition, which is vital for the success of checkpoint based therapy? A peptide derived from the protein p53 (residues 161–169 or p161) was reported to show this behavior. T cells recognizing this unmodified peptide could be further stimulated in vitro to create effective cancer killing CTLs (cytotoxic T lymphocytes). We hypothesize that the underlying difference may arise from post-translational glycosylation of p161 in normal individuals, likely masking it against recognition by TCR. Defects in glycosylation in cancer cells may allow the presentation of the native peptide. We investigate the structural consequences of such peptide glycosylation by investigating the associated structural dynamics.

## 1. Introduction

Major histocompatibility complex (MHC), or human leukocyte antigen in human (HLA) class I, belongs to a family of proteins that are found on the surfaces of cells and display endogenous peptides to T cell receptors (TCR) on cytotoxic T cells, resulting in the triggering of immune responses if the peptide displayed is recognized as a non-self-antigen [1]. These endogenous peptides are generated by the proteolysis of intracellular proteins [2], following which, the peptides form complexes with the MHC class I receptor proteins. The MHC class I or HLA class I proteins can be subdivided into three subtypes, namely HLA-A, HLA-B, and HLA-C, of which HLA-A24 (or A24) is the most abundant allele in Asian and Caucasian populations [3].

P53 is a tumor suppressor protein that guards cells against stress and its mutants are often found associated with multiple cancers [4,5]. Although p53 is thought to normally reside in the nucleus, missense mutations can result in its accumulation in the cytoplasm as well as in the nucleus. It is known that oncogenic mutations in p53 eliminate its tumor suppressor function, promote cell growth and proliferation, and inhibit apoptosis [6]. This suggests that mutant p53 peptides can be presented on MHC class I receptors and recognized as non-self, resulting in recognition as a non-self-antigen by appropriate TCRs, culminating in the elimination of those cells [7]. Yet, p53 mutant tumors develop, suggesting that certain factors must inhibit the process of rigorous immune surveillance. Could this result from lower levels of mutant p53 in cancer patients, which translate into insufficient levels of peptides expressed on MHC [8]? Or could this result from post-translational modifications in the peptide that is presented on MHC and mask its recognition by the immune system?

An intriguing recent observation [9,10] was the induction of p53 specific cytotoxic T lymphocytes upon incubation of PBMC’s from hepatocellular carcinoma and pancreatic adenocarcinoma A24-carrying patients with a synthetic p53-derived peptide (residues 161–169 of p53, sequence AIYKQSQHM, hereafter referred to as p161). Yet, this peptide p161 did not elicit an immune response in PBMC cells from healthy A24-carrying people. Why does the peptide trigger an immune response in patients, but not in healthy individuals despite the two peptides having identical sequences? We do not have any data to comment on the levels of peptides that are expressed and, hence, we decided to explore the putative consequences of post-translational modifications. Of the post-translational modifications known, the effects of glycosylation remain relatively unexplored. We wondered whether glycosylation could account for the differences between the effects of the p53 peptides, since cancer cells are known to have altered glycosylation [11]. Glycosylation of peptides and subsequent expression on MHC has been known [12], including disabling recognition by T cells, or the generation of new epitopes [13], giving rise to glyco-peptide specific immune responses in mice [14]. Glycosylation of a decapeptide from mouse hemoglobin was found to have resulted in changes to its binding affinity to TCR, depending on the location of the glycosylated residue [15]. Glycosylation of the Sendai virus nucleoprotein nonapeptide was found to elicit cytotoxic T lymphocytes [16]. Glycosylated human immunodeficiency virus (HIV) derived peptides were found to be masked against T cell recognition in some cases, and yet, were found to produce neo-epitopes that were recognized by T cells [17]. Direct recognition of glycan from the glycopeptide, representing an immunodominant epitope of Sendai virus by TCRs of CD8+ cells, was revealed by combining crystallography and modeling [18]. Clearly these studies suggest that glycosylation of epitopes can affect immune recognition, albeit in different ways. Nuclear proteins, such as p53, are known to be glycosylated, particularly with the monosaccharide *N*-Acetylglucosamine (GlcNAc) [19,20,21,22] and *N*-Acetylgalactosamine (or GalNAc) [23]. However, we could not find any reports on the role of glycosylation in the presentation and recognition of p53 peptide epitopes.

In this study, we develop atomistically detailed structural models of the p53 peptide p161 complexed to HLA-A24 and explore the consequences of the introduction of the two types of monosaccharides, GlcNAc and GalNAc, on its binding to HLA-A24. We also explore the consequences of glycosylation on the putative interactions that TCR makes with the A24-peptide complexes by simulating their structural dynamics. While we are making several assumptions here regarding the glycosylation of p161 and interactions with TCR, we hope to stimulate discussions on such possibilities and hope that experiments may illuminate new directions in vaccine development.

The region 161–169 corresponding to p161 lies in a solvent-exposed region of the crystal structures of the globular DNA binding domain (DBD) of p53 (Appendix A), suggesting that S166 could possibly get glycosylated, and is explored here. Y163C and Q167K are two mutants that are observed with high frequency in cancer patients [24,25,26], and lie in this region. It is of course possible that these mutations obstruct the attachment of the glycosyl groups at S166 and, hence, this is also explored. Next, we investigate how p161 (in unglycosylated and glycosylated states) may be accommodated in the peptide presentation pocket of MHC class I and what the consequences of these binary complexes are on their interactions with TCR.

## 2. Methods

### 2.1. Model Building

#### 2.1.1. Model of p53 and Its Glycosylation

The potential of p53 to be glycosylated was explored using the web-based predictor GLYCAM-Web [27] in the current study (this server has been used previously to successfully predict glycosylation sites of HIV-1 envelope [28]). The server requires the 3-dimensional structure of the protein of interest as input. We used the crystal structure of the wild type p53 tetramer bound to DNA (PDB ID: 2AHI, resolution 1.8 Å [29]). This structure consists of 4 chains of monomeric DBD arranged as a dimer of dimers (monomers ranged in length from residues 94–293) bound to two fragments of decameric DNA. For our calculations, we used residues 96–291 (this corresponded to the smallest resolved chain among the tetramers) of chain A.

The GLYCAM-Web server predicted that among the 31 residues in a monomer (14 Thr and 17 Ser) likely to be glycosylated (Table 1), ~58% or 18 residues can potentially be glycosylated (‘+’ symbol in the fifth column of Table 1). The residues S121 (GLYCAM score of 56.3), and S149 (GLYCAM score of 69.3) are both reported to be glycosylated and, hence, serve as benchmarks [21,30]. The score for S166 is 91.3, which is higher than that of 94% of all the residue scores. S121, S149, and S166 are all located in exposed loops, and hence, it is not surprising that they have similar predicted glycosylation propensities. Two commonly occurring mutations in this region are Y163C and Q167K. We modeled the mutations in the monomer. The GLYCAM score for S166 in the presence of the Y163C mutation remains the same, while the presence of Q167K results in a decrease of seven units (Table 1 and Appendix A). The GLYCAM score of S166 in the presence of Q167K is still higher than the GLYCAM score of the two controls, S121 and S149, both of which are known to be glycosylated. We conclude that the propensity of S166 to be glycosylated is high in the wild type p53 DBD and in the two mutants.

#### 2.1.2. Model of p161 Complexed to A24

Since the GLYCAM analysis suggested that S166 has a high probability of being glycosylated, we developed models of p161 (unglycosylated and glycosylated) in complex with A24. Analysis of the complexes between peptides and A24 showed that the B pocket of A24 is occupied by the sidechains of Y, F, M, or W amino acids from the peptides and the F pocket of A24 is occupied by the sidechains of F, L, I, W, or M amino acids from the peptides [3] (Figure 1). Guided by this, we hypothesized that the sidechains of residues Y163 and M169 of p161 should occupy pockets B and F of A24, respectively. In the absence of an available crystal structure of a complex between the nine amino acid long peptide p161 and A24, we constructed an atomic model of this complex using the crystal structure of A24 complexed to a nine amino acid (QFKDNVILL) peptide derived from the severe acute respiratory syndrome coronavirus nucleocapsid (hereafter referred to as SARS) protein (PDB ID: 3I6L, resolution 2.4 Å [31]), as a template.

We next set up and interrogated the structural dynamics of the following systems:A dimeric complex of A24 and an antigenic peptide (RFPLTFGWCF) from HIV and a trimeric complex of A24, peptide andTCR. Both these complexes were constructed from a crystal structure of the trimeric complex of A24 with the HIV peptide and TCR (PDB ID: 3VXU, resolution 2.7 Å [32]). The A24-HIV peptide dimer and the A24-HIV peptide-TCR trimer were simulated to examine the ability of our force field to generate stable complexes of these systems during the MD simulations;A dimeric complex of A24 and p161 and a trimeric complex of A24, p161 and TCR were constructed to examine the structural dynamics of p161 in complex with A24 and TCR. The dimer was constructed using the crystal structure of the complex between A24 and a SARS peptide (PDB ID: 3I6L, resolution 2.4 Å [31]), while the trimer was built using the crystal structure of the trimeric complex of A24-HIV-TCR (PDB ID: 3VXU, resolution 2.7 Å). The reason we did not build the dimer from the same structure as the trimer is because the peptide in the trimer is 10 amino acids long, while our p161 peptide is only 9 amino acids long;A dimeric complex of A24 and Galp161 (a monosaccharide GalNAc attached to the O^γ^ atom of the residue S166 in p161) and a trimeric complex of A24, Galp161 and TCR were constructed to examine the structural dynamics of the effects of glycosylation of p161 with GalNAc in complex with A24 and TCR;A dimeric complex of A24 and Glcp161 (a monosaccharide GlcNAc attached to the O^γ^ atom of the residue S166 in p161) and a trimeric complex of A24, Glcp161 and TCR were constructed to examine the structural dynamics of the effects of glycosylation of p161 with GlcNAc in complex with A24 and TCR.

TCRs are known to have the ability to bind to different MHCs displaying a variety of peptides [33]; this allows us to model the general binding mode and interaction with the A24-p161 complexes, with associated localized structural differences. All of the models were built using the program MODELLER v 9.10 [34]. Ten models were constructed and the model with the lowest DOPE (discrete optimized protein energy) [35] score was chosen. 

### 2.2. Molecular Dynamics (MD) Simulations

The constructed atomic models were next subjected to molecular dynamics (MD) simulations. The protonation states of ionizable residues were determined using the reduce method [36] in the MolProbity server [37]. The *N*- and *C*-termini of the protein segments were capped using acetyl (ACE) and *N*-methyl (NME) groups respectively; to simulate the experimental conditions [10], we did not cap the peptide. The monosaccharide was added using the GLYCAM server [38].

The program AMBER16 [39] with the ff14SB force field [40] was used to model the proteins and peptides, while the Glycam06 force field [38] was used to model the attached monosaccharides. All the complexes were solvated with TIP3P water molecules [41] in a periodic truncated octahedron box, and the net charge was neutralized with either chloride or sodium ions. The distance between any edge of the box and any protein atom was at least 10 Å. The temperature was kept constant with a collision frequency of 2 ps^−1^ using a Langevin thermostat [42]. A constant pressure of 1 atm was maintained with a Berendsen barostat with a relaxation time of 2 ps [43]. The Particle Mesh Ewald method [44] was used to calculate long-range electrostatic interactions for interatomic distances greater than 9 Å. The Settle [45] algorithm was used to constrain bond vibrations involving hydrogen atoms, which allowed a time step of 2 fs during the simulations. Coordinates were saved at an interval of 100 ps. The systems were first minimized for 5000 steps using the steepest descent algorithm followed by 5000 steps of conjugate gradient algorithm with weak harmonic positional restraint of 2 kcal mol^−1^ Å^−2^ applied to the heavy atoms. The systems were then gradually heated at constant volume to 300 K for 50 ps followed by equilibration at a constant pressure of 1 atm for another 50 ps. Subsequently, 2 ns of unrestrained MD simulations for equilibration was applied to each system, and finally, production runs for 500 ns at 1 atm and 300 K were carried out. RMSD plots were used to check the convergence of the simulations. All simulations were run in triplicate. The MD simulations for the tertiary complexes were extended to 1 μs. Details of the simulation setup for each system are shown in Table 2.

The enthalpic components of the free energies (and decomposition) of binding between the molecules were estimated using 200 equally spaced snapshots from the simulations, using the Molecular Mechanics/Generalized Born Surface Area (MM/GBSA) method, as described earlier [46,47]. Despite its approximations, the method has successfully been used to gain insights, particularly when comparing systems [48,49].

Hydrogen bond (hbond) analysis was carried out using the cpptraj command in AMBER16; only hbonds with a distance of less than 3.5 Å between the acceptor and donor heavy atoms, and with a bond angle of at least 135° were considered.

## 3. Results

### 3.1. Benchmarks with the Complex of A24 and the HIV Peptide

The C^α^ RMSD (Appendix A) of the complex between A24 and HIV in the presence and absence of TCR suggests that the simulations are stable in the chosen force field. Nine snapshots taken at 300, 400, and 500 ns from each of the triplicate MD simulations of the A24-HIV complex and nine snapshots taken at 800, 900, and 1000 ns from each of the triplicate MD simulations of the A24-HIV-TCR complex are shown (Appendix A), and suggest that the atomic movements are stable with no large-scale motions. Similar motions have been reported from computational and experimental analyses of MHC Class I bound to a different set of peptides [50]. We chose the last 200 ns of the MD simulations of the A24-HIV and A24-HIV-TCR complexes for further analysis.

The root mean square fluctuations (RMSFs) of HIV, A24, and TCR (Figure 2) over the last 200 ns of the MD simulations show that the peptide fluctuates between 0.5–2 Å. The RMSF trends of A24 are similar between the two complexes and are lower than 4 Å, with most large-scale motions arising from loops. The RMSF values of TCR are lower than 3 Å. These observations are similar to those reported from MD simulations of the complexes between peptides and MHC class I in the presence and absence of TCR [51,52]. These findings suggest that the ff14SB force field used in our study produces stable trajectories during the MD simulations of the complexes studied here and, hence, is appropriate for our investigations.

### 3.2. Structural Dynamics of A24 Complexed to the p161 Peptides

The C^α^ RMSD (Appendix A) of the A24-peptide complexes, where the peptides are p161, Galp161, and Glcp161, are stable with values ranging from 1 to 4 Å. The RMSD values of the peptides are attenuated upon glycosylation, while A24 undergoes larger deviations from the starting states upon glycosylation; this presumably results from the protein conformational changes to accommodate the sugar moieties. The nine snapshots (taken at 300, 400, and 500 ns from each of the triplicate MD simulations) show that the peptides are stable in their binding pockets (Appendix A). The RMSD values have converged by 200 ns and hence, the 300–500 ns region is chosen for analysis.

The fluctuations of A24 in the three A24-peptide complexes are largely similar to each other and are similar to the values seen in the A24-HIV complex (Appendix A), with low values in the peptide-binding regions and high values in the loops. However, there are some differences in the flexibilities of the *N*- and *C*-termini of the peptides and their surrounding environments. The motions seen in A24-p161 are slightly larger than in the glycosylated complexes. This attenuation is expected as the sugars engage in local polar interactions with the protein atoms.

#### 3.2.1. Hydrogen Bond (Hbond) Interactions

The interactions between the p161 peptides and A24 are mediated by several hbonds. The most stable hydrogen bonds (lifetime > 80% during the last 200 ns of the MD simulations) involve the backbone atoms of p2, p7, p8, and p9 and the sidechain atoms of p3 and p4. These interactions are conserved regardless of the glycosylation status. Two hbonds (lifetimes~60%) that exist between the backbone of residue M9 in p161 and the sidechains of Y84 and T143 in A24 are lost when p161 is glycosylated. An hbond between the sidechain of S6 in p161 and the sidechain of D74 in A24 is weak (lifetime~36%) and is replaced by new hbond interactions formed between the sugar atoms and the sidechain of residue T73 (lifetime~36–50%) or the backbone of residue K66 (lifetime~23%). In the presence of the sugars, the hbond interaction between the backbone atom of Y3 in the peptide and the sidechain atom of E63 in A24 is strengthened by ~50% (lifetimes of hbonds are 43, 90, and 99% for p161, Galp161, and Glcp161, respectively); the hbond variations across the 3 systems are shown in Figure 3.

#### 3.2.2. Energetics of Interactions of Peptides and A24

The computed energies of binding of the three peptides to A24 (Table 3) suggest that the presence of the sugars results in reductions in the binding energies by ~8 kcal mol^−1^.

The decomposition of the total energy of binding into contributions from residues shows that most of the energy arises from interactions of the sidechains of the peptide with A24. The sidechains that occupy the two main recognition pockets (B and F pockets) i.e., Y3 and M9 of p161 make the largest contributions (~7 kcal mol^−1^) to the total energy (Figure 4 and Appendix A). In addition, the residues I2, K4, and Q7 contribute >2 kcal mol^−1^. These contributions remain the same even upon glycosylation; the only differences appear to be in the interactions of the glycosylated residue S6 and neighboring residue Q5, each undergoing a weakening of ~1 kcal mol^−1^. However, the loss is compensated for by the attached monosaccharide. Although no major differences are apparent across the three peptides, analysis of the total contributions from both sidechains and backbone atoms show that I2 and Y3 of the glycosylated peptides have stronger contributions from I2 and Y3 (~1–2 kcal mol^−1^) and weaker contributions (~1–2 kcal mol^−1^) from H8 and M9. In general, it appears that glycosylation results in strengthening the interactions at the *N*-terminus (blue in Figure 4) and weaken the interactions at the *C*-terminus (red in Figure 4).

The next question that arises is: does glycosylation and associated changes affect the recognition of this complex by TCR and trigger the immune system? While we are unable to explore the whole process of the triggering of an immune response, we apply MD simulations to explore the consequences upon glycosylation for the ternary complexes of A24-peptides-TCR.

### 3.3. The Interactions of TCR with the A24-Peptide Complexes

#### 3.3.1. RMSD and RMSF

The C^α^ RMSD values of the individual molecules in the A24-peptide-TCR complexes range between ~1–4 Å (Appendix A); the RMSD of the peptide alone is <2 Å. The RMSD of TCR and A24 are slightly higher (~3 Å). The RMSD values of both A24 and peptides are similar to those sampled in the binary A24-peptide complexes. In general, the RMSD trends suggest that the dynamics are stable after 600 ns and so we use the 800–1000 ns period for analysis.

The RMSF trends of A24 residues in the ternary A24-peptide-TCR complexes are similar to each other and are similar to the values seen in the MD simulations of the benchmark A24-HIV-TCR complex (Appendix A). While the residues at the interaction sites fluctuate < 2 Å, the residues at the loop regions, unsurprisingly, have higher fluctuations. The RMSF of p161 and its glycosylated analogs are minimal (0.5–1 Å). In the TCR, the residues that make up the regions that interact with A24 and/or with the peptides fluctuate < 2 Å (Appendix A), while residues that have higher fluctuations are located mostly in the loop regions. The fluctuation trends of TCR in the MD simulations of the three ternary A24-peptide-TCR complexes are similar.

#### 3.3.2. Binding Energy

The binding energies between the peptides and A24 follow the same trend in the presence of TCR as in the MD simulations of the binary complexes (Table 3 and Table 4), i.e., glycosylation attenuates the interactions with A24 also in the presence of TCR, by ~4 kcal mol^−1^. However, glycosylation strengthens the interactions between the peptide and TCR by ~5–6 kcal mol^−1^. The binding energy of A24-peptide to TCR is weakened by ~2.5 kcal mol^−1^ in the presence of GlcNAc and strengthened by ~13 kcal mol^−1^ in the presence of GalNAc.

#### 3.3.3. Hbond Interactions between A24 and Peptide

The backbone atoms of I2, Q7, H8, and M9 make five stable hbonds with A24 (lifetime > 80%) in the MD simulations of the ternary A24-peptide-TCR complexes (Appendix A). These hbonds are seen in both the MD simulations of the binary complexes of A24-peptides and the ternary complex of A24-peptide-TCR. A stable hbond between Y3 and A24 seen in the binary complexes of A24-Galp161 (lifetime~90%) is weakened in the ternary complex of A24-Galp161-TCR (lifetime~34%). Similar to the A24-peptide complex, in the A24-peptide-TCR complexes, this hbond between the backbone of Y3 and sidechains at E63 is weak (lifetimes~43% and ~54%, respectively) if the peptide is p161, and strong (lifetimes~99% and ~98%, respectively) if the peptide is Glcp161. The presence of TCR strengthens the hbonds between the sidechain of S6 and D74 of A24 (lifetime increases from ~30% to ~60%). The hbonds between the backbone of M9 with the sidechains of Y84 and T143 is observed only in the A24-p161, but not its glycosylated complexes and is now also observed both in A24-p161-TCR and A24-Galp161-TCR complexes. These hbond interactions are a major determinant of the better binding energetics of p161 to A24 compared to the glycosylated peptides in the MD simulations of the binary complexes. 

#### 3.3.4. Hbond Interactions between Peptide and TCR

The strongest interactions between the peptide and TCR in the MD simulations of the ternary A24-peptide-TCR complexes is with Glcp161, followed by Galp161, and the weakest is the unglycosylated p161. The hbond interactions between the peptide and TCR mirror this trend and is not surprising, given the number of polar groups that the glycosylated peptide presents to the TCR (Appendix A). One interesting observation is that the difference arises despite the only difference between GlcNAc and GalNAc being in the orientation of the hydroxyl group at position 4 (either equatorial or axial). How might such a small difference result in the differences in binding energetics? The axial positioning of the hydroxyl in GalNAc brings it close to H8, as the hydroxyl groups at positions 3, 4, and 6 can now interact with H8. In GlcNAc, this hydroxyl cannot interact with H8 and instead interacts with TCR.

#### 3.3.5. Hbond Interactions between A24 and TCR

Among the three A24-peptide-TCR complexes (Appendix A), if the peptide is Galp161, the hbond and salt bridge interactions between A24 and TCR have the longest lifetimes, followed by the case where the peptide is p161; the least stable interactions between A24 and TCR occur when the peptide is Glcp161 (Appendix A). These hbond and salt bridge interactions are consistent with the computed binding energies.

### 3.4. The Proximity of TCR to the A24-Peptide Complexes

We next explore whether glycosylation affects the proximity of TCR to the A24-peptide complexes. We compute the distance between the complementarity-determining regions (hereafter referred to as CDR) of TCR and α/β helix of A24. Both CDR of αTCR and βTCR include three regions, namely, CDR1α, CDR2α, CDR3α, CDR1β, CDR2β, and CDR3β ranging from α25–32, α51–53, α92–99, β26–30, β50–56, and β95–99, respectively. The distance between CDRα and βA24 or CDRβ and αA24 (Figure 5) suggest that the TCR is “farthest” from the A24-peptide upon glycosylation with GlcNAc; this is also seen in snapshots of the three systems taken from the MD trajectory (Figure 6).

At this point, the observations based on our simulations (up to 3 μs for each ternary complex) led us to a cautious conclusion that the lack of TCR response to the presence of the p53 peptide on the MHC likely results from the glycosylation of the peptide, specifically with GlcNAc, whose presence clearly appears to potentially abrogate the TCR recognition. This is also evident in the interactions between the A24-peptide complexes, with TCR being the weakest (Table 4).

## 4. Discussion

The process of discrimination of the immune system between the self and non-self is a highly efficient, but complex process, dependent on sensing by α/β TCR of antigenic peptides, expressed by the MHC proteins on the surfaces of cells [53]. Yet, diseases, such as autoimmune disorders and cancers, result from the inability of the immune system to appropriately respond, suggesting that this complex system is prone to frailties. What is the nature of the biomolecular mechanisms that underpin these processes? For example, how is it that cells carrying mutant forms of say a protein, such as p53 or RAS, which are known to be oncogenic, survive? This could result from the expression of low levels of the peptide carrying the mutation, which does not reach a minimal concentration that may be required to activate the immune system. Recently, this hypothesis was exploited and antibodies were successfully raised against peptidic antigens corresponding to R175H, a common p53 mutation, expressed on a common HLA allele [8] and against peptidic antigens corresponding to two common RAS mutations, G12V and Q61H/L/R [54]. On the other hand, the MHC may express post-translationally modified peptides, which could either activate the immune system with its neo-glyco-epitopes, or indeed mask the peptides from recognition [17].

In this current study, we speculated that a post-translational modification, glycosylation, of a p53 peptide might result in the loss of its recognition by TCR. The current study was inspired by reports [9,10] of the induction of cytotoxic T lymphocytes directed to a synthetic p53-derived peptide (p161 corresponding to residues 161–169 of p53, sequence AIYKQSQHM) in PBMC derived from hepatocellular carcinoma and pancreatic adenocarcinoma A24-carrying patients. Yet, this peptide p161 did not elicit an immune response in PBMC from healthy A24-carrying people (Figure 7). We speculated that this peptide is presented by MHC molecules in a glycosylated form, which prevents interactions with TCR in healthy people. We explored the attachment of two common glycosylations, GalNAc and GlcNAc, at the predicted site for glycosylation, S166. Our molecular dynamics simulations suggest that both modified versions of the peptide bind equally strongly to MHC. The difference in location of the hydroxyl group at position 4 in GalNAc and GlcNAc does not really cause any significant difference in interactions with A24. However, the interaction of TCR, with the GalNAc modified peptide resulted in stronger interactions compared to A24 with the unglycosylated peptide. In contrast, when the peptide is modified by GlcNAc, the associated small shift in the hydroxyl group at the 4 position of the added sugar resulted in a dramatically weaker interaction with TCR, and compels us to speculate that the peptide is modified by GlcNAc. This may mask the A24-peptide complex from proper recognition by TCR. There is also some evidence that increased stability of the MHC-peptide complex and increased residence times of the peptide on MHC may be required for triggering an immune response [55], and it is possible that this plays a role in the lack of response in healthy individuals in this particular case. However, the role of glycosylation remains complex. Glycosylation of epitopes in a virus has been shown to prevent T cell recognition and, hence, immune escape of a virus epitope [56]. On the other hand, GlcNAc modified peptides can create new epitopes and have been shown to elicit glycopeptide specific immune responses in mice [14]. Our model suggests that cancer cells process exogenous p161 and present it on MHC without glycosylation, thus triggering T cell recognition and subsequent processes. In contrast, in healthy cells, the same exogenous peptide is glycosylated, presented on MHC, and does not trigger T cell recognition. These observations raise several questions. What pathways (Figure 7) determine the presentation of the peptides and are they different between cancer and normal cells? It is known that wild type p53 increases the presentation of peptides on MHC while mutant p53 (characteristic in pancreatic cancer cells, the ones used in the study simulated here) can dysregulate it [57,58]. It is also known that cancer cells can evade immune recognition through losing MHC I antigen presentation [59]. If the administration of synthetic peptide p161 results in it being sensed by the cells as an exogenous peptide, then is it necessarily only expressed on MHC II [60,61]? Maybe the exogenous peptides are expressed without glycosylation on MHC II in both cancer and healthy cells, but the density of MHC I in p53 wild type containing healthy cells is much higher [58], and only the MHC I presented peptides are glycosylated, resulting in abrogation of T cell recognition. There is also evidence that exogenous molecules can actually traverse the cell entrapped in vesicles [62] and, hence, are protected from glycosylation, and it is possible that the pathways mediating the journey of the exogenous peptide to the ER are different between cancerous and healthy cells. Classical models suggest that the endogenous version of p161 will most probably be glycosylated prior to entry into the ER through TAP, loaded on to MHC I, and then expressed on the cell surface. But, there is evidence that glycosylation pathways are altered in cancers [11]. Tagging the peptide with a spectroscopic probe, followed by high resolution mapping of its path, may shed light on this. Nevertheless, the models presented in this study suggest a plausible mechanistic explanation for the lack of TCR response to exogenously administered peptide p161 to cells from healthy people, while eliciting a TCR response in cells from cancer patients. If validated, such a mechanism may add to understanding the complexity of discrimination between self and non-self by the immune system [63].

## Figures and Tables

**Figure 1 biomolecules-11-01056-f001:**
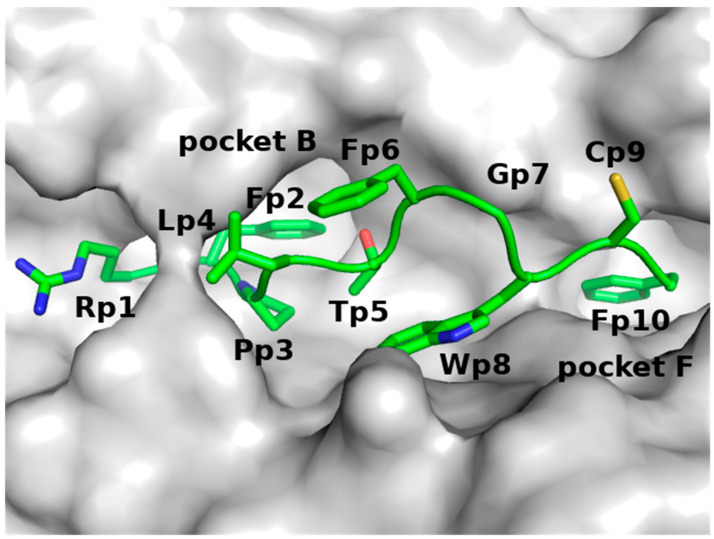
The pocket of A24 in complex with the HIV peptide, taken from the crystal structure of the A24-HIV-TCR complex (PDB ID: 3VXU, resolution 2.7 Å), where A24 is in grey surface and the HIV peptide is in green sticks with Oxygen atoms in red and Nitrogen atoms in blue.

**Figure 2 biomolecules-11-01056-f002:**
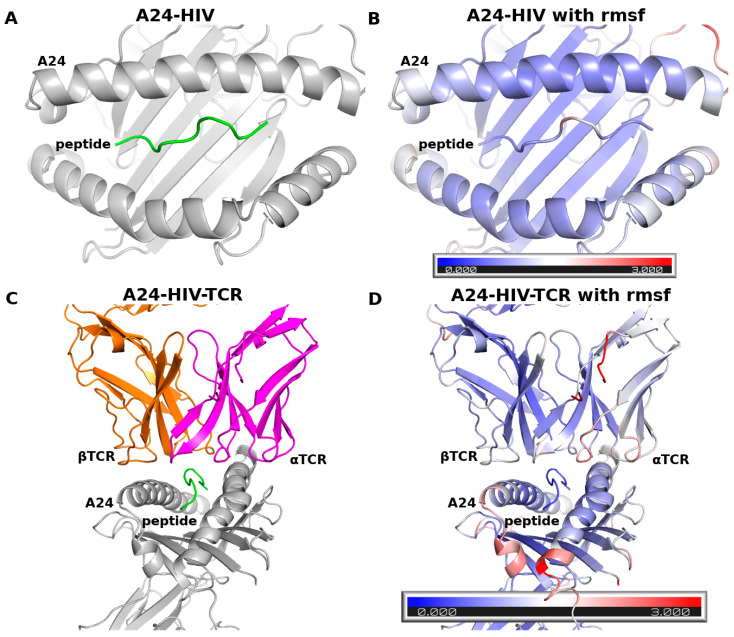
C^α^ RMSF during the last 200 ns of the MD simulations mapped onto the structures of (**A**,**B**) A24-HIV and (**C**,**D**) A24-HIV-TCR complexes; the colors represent the fluctuations from very low (blue) to 3 Å (red).

**Figure 3 biomolecules-11-01056-f003:**
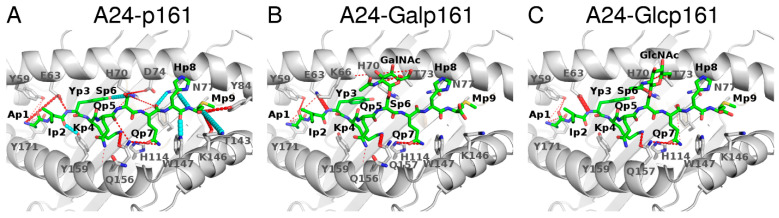
Hbond interactions between A24 and the p161 peptides during the 300–500 ns part of the MD simulations: (**A**) p161, (**B**) Galp161, and (**C**) Glcp161. A24 is in grey ribbons, the interacting residues from A24 are in sticks, the peptides are in green sticks with Oxygen atoms in red and Nitrogen atoms in blue. Hbonds that are present in all three complexes with a lifetime > 10% are shown as dashed cyan lines in only A24-p161 and not shown in the other panels for clarity. If an hbond has a lifetime difference between a glycosylated complex and a wild type complex higher than 10%, then that hbond is shown in dashed red lines. The thickness of the dashed lines corresponds to the lifetime. In the case of multiple hbond donors and acceptors, as in Arg and Asp, the atoms between the donor/acceptor atoms are used to show the hbonds.

**Figure 4 biomolecules-11-01056-f004:**
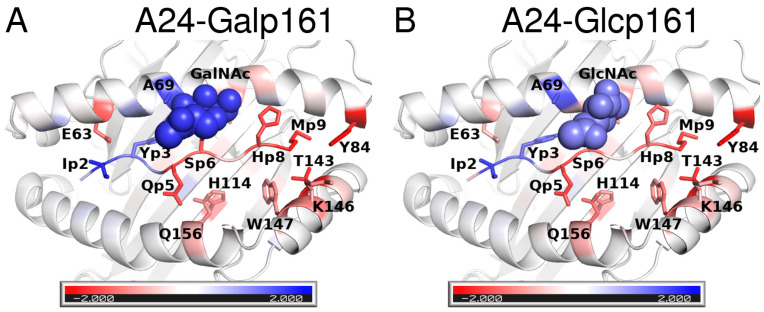
Differences in contributions of the peptide residues to the total interaction energies with A24. The difference is between the interactions of p161 and those of the glycosylated peptides, namely (**A**) Galp161 and (**B**) A24-Glcp161. The colors correspond to energies ranging from −2 kcal mol^−1^ to 2 kcal mol^−1^ where red indicates that p161 residues are more stabilizing, while blue indicates that p161 residues are destabilizing. Residues that undergo large differences upon glycosylation are shown in sticks. GalNAc and GlcNAc are shown in spheres.

**Figure 5 biomolecules-11-01056-f005:**
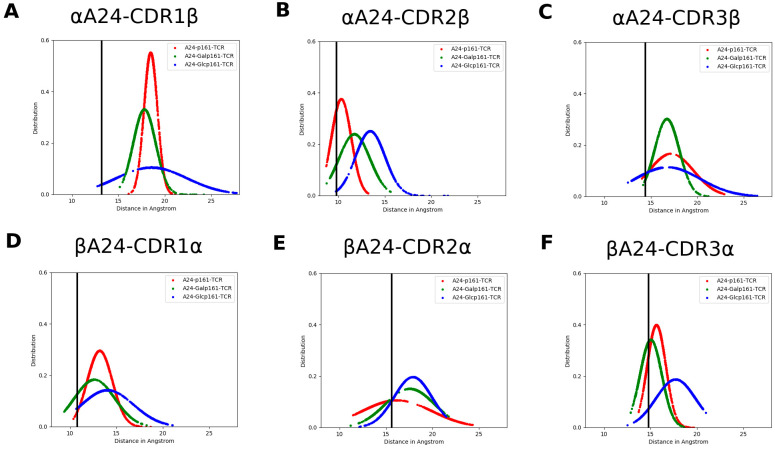
The distance distribution between A24 and TCR, namely (**A**) αA24 and CDR1β, (**B**) αA24 and CDR2β, (**C**) αA24 and CDR3β, (**D**) βA24 and CDR1α, (**E**) βA24 and CDR2α, and (**F**) βA24 and CDR3α in the MD simulations of the A24-peptide-TCR complexes where the peptides are p161 (red), Galp161 (green), and Glcp161 (blue). The black line is the corresponding distance in the crystal structure of the A24-HIV-TCR complex (PDB ID: 3VXU).

**Figure 6 biomolecules-11-01056-f006:**
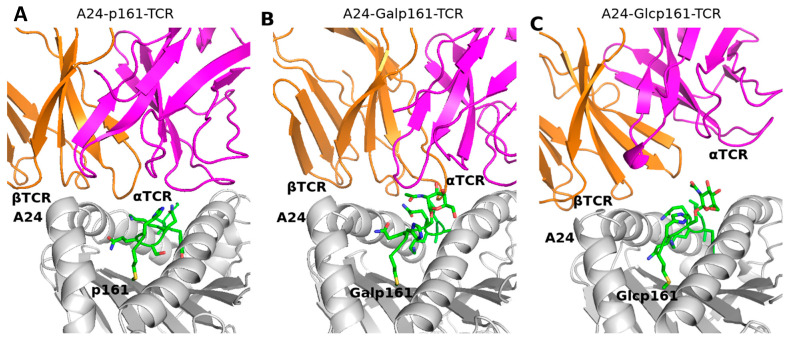
Snapshots taken from the MD simulations of (**A**) A24-p161-TCR, (**B**) A24-Galp161-TCR, and (**C**) A24-Glcp161-TCR complexes. A24, peptide, α- and β-chains of TCR are in grey, green, pink, and orange ribbons, respectively. The interaction residues are shown in sticks with Oxygen atoms in red and Nitrogen atoms in blue.

**Figure 7 biomolecules-11-01056-f007:**
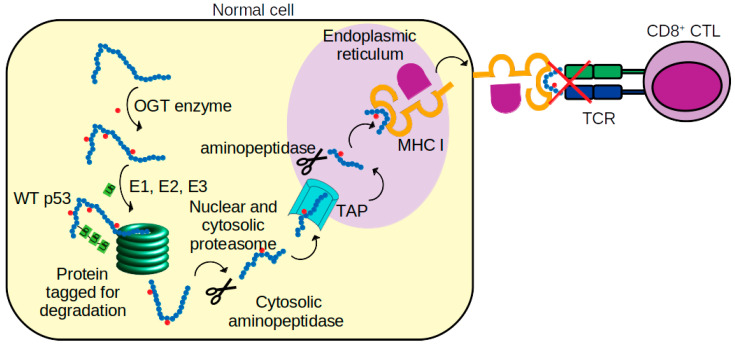
Schematic of p53 peptide presentations to cytotoxic T lymphocytes (CTL). p53 protein is first glycosylated (small red filled circles) by O-GlcNAc transferase (OGT) [64]. The glycosylated p53 is then tagged by ubiquitin (Ub chains) by ubiquitin-activating (E1), ubiquitin-conjugating (E2), and ubiquitin-protein ligase (E3) enzymes. Next, the protein is digested into peptides by proteasomes and aminopeptidase enzymes. The peptides are later translocated, via transporters associated with antigen processing (TAP), into the endoplasmic reticulum (ER). In the ER, the peptides are again trimmed by aminopeptidase and loaded on to nascent MHC class I molecules [65]. MHC class I-peptide complexes are then translocated to the surface of the tumor cell and recognized by the T cell receptor (TCR) of CD8^+^ CTL cell. Our simulations suggest that in some normal cells, the glycosylated p53 peptide in the complex with MHC I is not recognized by TCR.

**Table 1 biomolecules-11-01056-t001:** The propensity for glycosylation of monomeric DBD of p53_96-291_ taken from the crystal structure of a dimer of dimers of DBD bound to two decamers of DNA (PDB ID: 2AHI, resolution 1.8 Å, [29]). The first column is the sequence from residues i − 2 to i + 2 (residue i is marked with *); the *N*-terminus sequence is from i to i + 2 or i − 1 to i + 2. The second column is the identity of the chain in the pdb file. The third column is the residue number (i) of the Thr or Ser residue. The fourth column is the GLYCAM score of the particular Ser or Thr in monomeric DBD while the fifth column lists the glycosylation propensity of the residue, i.e., positive (+), zero (0), and negative (−), corresponding to high, low, and zero propensity for glycosylation. The sixth column is the difference between the GLYCAM score of the monomeric wild type DBD and the monomeric Y163C mutant DBD. The seventh column is the difference between the GLYCAM score of the monomeric wild type DBD and the monomeric Q167K mutant DBD.

Residue Name	Chain	Residue Number	GlyCAM Score for Wild Type Monomeric DBD	Glycosylation Status	Difference Compared to Wild Type Monomeric DBD	
					Y163C	Q167K
S*VP	A	96	99.8	+	0	0
VPS*QK	A	99	57.7	+	0	0
QKT*YQ	A	102	51.9	+	0	0
QGS*YG	A	106	89.2	+	0	0
LHS*GT	A	116	31.5	0	0	0
SGT*AK	A	118	19.4	−	0	0
AKS*VT	A	121	56.3	+	0	0
SVT*CT	A	123	56.8	+	0	0
TCT*YS	A	125	1.3	−	0	0
TYS*PA	A	127	0	−	0	0
AKT*CP	A	140	41.6	+	0	0
VDS*TP	A	149	69.3	+	0	0
DST*PP	A	150	63.2	+	0	0
PGT*RV	A	155	1.3	−	0	0
KQS*QH	A	166	91.3	+	0	6.9
HMT*EV	A	170	83.9	+	0	0
RCS*DS	A	183	111.7	+	0	0
SDS*DG	A	185	68.3	+	0	0
RNT*FR	A	211	37.2	0	0	0
RHS*VV	A	215	5.2	−	0	0
VGS*DC	A	227	41	+	0	0
DCT*TI	A	230	17.5	−	0	0
CTT*IH	A	231	35.5	0	0	0
CNS*SC	A	240	13.4	−	0	0
NSS*CM	A	241	62.2	+	0	0
ILT*II	A	253	0.7	−	0	0
IIT*LE	A	256	14.5	−	0	0
EDS*SG	A	260	70.4	+	0	0
DSS*GN	A	261	81.2	+	0	0
RNS*FE	A	269	14.7	−	0	0
RRT*EE	A	284	70.8	+	0	0

**Table 2 biomolecules-11-01056-t002:** The A24-peptide complexes in the presence and absence of both kinds of glycosylation and TCR subjected to MD simulations (in triplicate).

Protein	Peptide	TCR	Number of Waters	Total Number of Atoms	Simulation Time (μs)
A24	HIV	−	14,130	48,538	0.5 * 3 = 1.5
A24	p161	−	14,746	50,364	0.5 * 3 = 1.5
A24	Galp161	−	14,742	50,379	0.5 * 3 = 1.5
A24	Glcp161	−	14,736	50,361	0.5 * 3 = 1.5
A24	HIV	TCR	64,105	205,288	1.0 * 3 = 3.0
A24	p161	TCR	64,673	206,970	1.0 * 3 = 3.0
A24	Galp161	TCR	64,705	207,093	1.0 * 3 = 3.0
A24	Glcp161	TCR	64,675	207,003	1.0 * 3 = 3.0

**Table 3 biomolecules-11-01056-t003:** Binding energies ΔH_GBSA_ between the p161 peptides and A24 during the 300–500 ns phase of the MD simulations with the standard deviations in parentheses.

Energy	A24-p161	A24-Galp161	A24-Glcp161
ELE	−448.3 (11.0)	−400.3 (18.6)	−397.0 (6.8)
VDW	−97.5 (3.0)	−102.3 (0.8)	−99.5 (1.1)
GBSOL	459.8 (11.5)	427.6 (17.6)	421.8 (4.4)
ΔH	−86.0 (4.0)	−75.0 (0.7)	−74.6 (1.7)

**Table 4 biomolecules-11-01056-t004:** Binding energies ΔH_GBSA_ during the last 200 ns of the MD simulations of the ternary A24-peptide-TCR complexes. All the values are in kcal mol^−1^ and the standard deviations are in parentheses.

Interaction	Energy	A24-p161-TCR	A24-Galp161-TCR	A24-Glcp161-TCR
A24 vs. pep	ELE	−414.0 (10.5)	−427.9 (27.4)	−410.4 (3.9)
	VDW	−96.5 (0.4)	−91.6 (7.6)	−101.4 (3.3)
	GBSOL	430.3 (5.6)	443.4 (14.1)	436.1 (4.8)
	ΔH	−80.2 (6.1)	−76.1 (8.6)	−75.7 (1.9)
pep vs. TCR	ELE	−74.1 (12.2)	−79.5 (3.1)	−63.9 (5.2)
	VDW	−18.1 (3.7)	−26.9 (2.4)	−27.5 (10.2)
	GBSOL	87.5 (12.2)	96.7 (3.4)	80.1 (1.5)
	ΔH	−4.7 (2.4)	−9.7 (1.6)	−11.3 (6.1)
A24-pep vs. TCR	ELE	209.8 (44.9)	171.4 (43.9)	224.1 (60.4)
	VDW	−91.0 (5.8)	−99.4 (4.6)	−84.2 (17.9)
	GBSOL	−145.4 (41.4)	−111.5 (40.0)	−163.8 (59.6)
	ΔH	−26.5 (3.3)	−39.5 (8.6)	−23.9 (15.5)
A24 vs. TCR	ELE	284.0 (42.9)	250.9 (44.2)	288.0 (65.6)
	VDW	−72.9 (2.6)	−72.5 (4.5)	−56.8 (7.7)
	GBSOL	−228.6 (40.4)	−203.2 (40.7)	−239.6 (60.5)
	ΔH	−17.6 (2.3)	−24.8 (8.4)	−8.4 (10.9)

## Data Availability

The data presented in this study are available in the Appendix A. The input files for the MD simulations, and the analysis script is available at https://github.com/nguyenbinhchem/MD_p53_161_glycosylation (accessed on 28 June 2021).

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
