# Peer review of "Can Glycosylation Mask the Detection of MHC Expressing p53 Peptides by T Cell Receptors?"

_biomolecules, 2021, doi:10.3390/biom11071056_

Round 1
Reviewer 1 Report
This is an interesting theoretical study aiming to provide new mechanistic insights into how the immune system recognizes self and non-self agents, and how the proper immune response can be avoided due to pathological mutations, such as oncogenic mutations in TP53 protein. Specifically, the authors computationally evaluate the role of glycosylation in the interaction between human leukocyte antigen (HLA-A24) and a peptide derived from the major tumor suppressor gene TP53.
Technically the study demonstrates sufficient quality. The computational methods are appropriate, the data is adequately discussed and conclusive. The lack of experimental validation does not allow to fully evaluate the predictions, though this is clearly out of scope of this research.
I have two general suggestions. It would be helpful to include a schematic diagram explaining the general idea of the study on the pathway level: show where is TP53, where is A24, where the peptide, how glycosylation is important, what is the cellular response.
Related to the first comment, I found it difficult to follow the introduction and discussion. If possible I would suggest to rewrite introduction and Discussion to improve the clarity, focus on this study, and to provide additional premise to support the author's hypothesis.
Instead of listing many questions and possibilities on page 2 lines 50-69, it would be more helpful to provide known examples where PTM changed the self non-self recognition. Is there any experimental evidence that glycosylation or other PTMs of p161 are involved in immune response?
Discussion, line 400: “A key question is why is it that cells carrying mutant 400 forms of say a protein such as p53 or RAS, which are known to be oncogenic, survive?”. A short answer is that oncogenic mutations in TP53 eliminate its tumor suppressor activity, release the breaks from cell cycle, promote cell growth and proliferation, and inhibit apoptosis. I think authors could rewrite the question in the beginning of the paper and in the discussion to make it more focus, more related to this particular study, and to better orient the reader.
Author Response
Thank you for the referee comments on our article titled “Can glycosylation mask the detection of MHC expressing p53 peptides by T-cell receptors?” by Thanh Binh Nguyen, David P. Lane, Chandra S. Verma (Manuscript no 1245424) that we had submitted for consideration for publication in the special issue of Biomolecules on “Protein–Protein Interactions: Methods and Applications”.
We are now submitting our revised version with a point-by-point response to the referee’s suggestions.
We have rewritten a significant part of the Introduction and Discussions as requested by Referee 1 and added a Figure (Figure 7 in Discussions).

Reviewer 2 Report
In this paper Nguyen et al investigates if glycolysation of p53, p161, is reduced in cancer and thus aids antigen presentation in these patients.
The work is entirely computational. I have some concerns regarding the predictions made, which could beg for experimental validation normally. However, I assume this is out of scope. Instead, I am satisfied with the rather good description of methods and QC made.
Major points:
* The input files for the MD simulations should be available for reproducibility
* Scripts of any higher level of complexity should be put on github
Minor points:
* “Supplementary Materials: The following are available online.” I am not sure what this refers to
Author Response
Thank you for the referee comments on our article titled “Can glycosylation mask the detection of MHC expressing p53 peptides by T-cell receptors?” by Thanh Binh Nguyen, David P. Lane, Chandra S. Verma (Manuscript no 1245424) that we had submitted for consideration for publication in the special issue of Biomolecules on “Protein–Protein Interactions: Methods and Applications”.
We are now submitting our revised version with a point-by-point response to the referee’s suggestions.
We have expanded on the Supplementary material listed in the main manuscript together with putting the input files and methods of analysis in Github.

Round 2
Reviewer 2 Report
The new supplementary material resolves my concerns. Some copyediting may still be required (e.g. double spaces, similar)